# Towards synthetic cells using peptide-based reaction compartments

Kilian Vogele[1], Thomas Frank[1], Lukas Gasser[1], Marisa A. Goetzfried[1], Mathias W. Hackl [2], Stephan A. Sieber [2], Friedrich C. Simmel [1,3] & Tobias Pirzer [1]

Membrane compartmentalization and growth are central aspects of living cells, and are thus encoded in every cell's genome. For the creation of artificial cellular systems, genetic information and production of membrane building blocks will need to be coupled in a similar manner. However, natural biochemical reaction networks and membrane building blocks are notoriously difficult to implement in vitro. Here, we utilized amphiphilic elastin-like peptides (ELP) to create self-assembled vesicular structures of about 200 nm diameter. In order to genetically encode the growth of these vesicles, we encapsulate a cell-free transcription-translation system together with the DNA template inside the peptide vesicles. We show in vesiculo production of a functioning fluorescent RNA aptamer and a fluorescent protein. Furthermore, we implement in situ expression of the membrane peptide itself and finally demonstrate autonomous vesicle growth due to the incorporation of this ELP into the membrane.

[1] Physics of Synthetic Biological Systems-E14, Physics-Department and ZNN, Technische Universität München, 85748 Garching, Germany. [2] Department of Chemistry, Center for Integrated Protein Science Munich (CIPSM), Technische Universität München, Lichtenbergstraße 4, 85748 Garching, Germany. [3] Nanosystems Initiative Munich, 80539 Munich, Germany. Correspondence and requests for materials should be addressed to T.P. (email: pirzer@tum.de)

Life is based on the complex interaction of numerous molecular components, which self-assemble and self-organize into higher ordered structures. Inspired by natural systems, several aspects of living cells have already been recapitulated in vitro, such as DNA replication, protein expression or compartmentalization of molecular reactions[1,2]. Compartmentalized protein expression and DNA replication inside vesicles have been studied as well[3,4]. For bottom up approaches towards the creation of protocellular compartments, these aspects need to be coupled and coordinated.

For instance, DNA amplification was coupled with vesicle self-reproduction[3], or the genetically encoded synthesis of phospholipids inside lipid vesicles from precursor molecules such as acyl-coenzyme A and glycerol-3-phosphate[5]. Here, mainly phospholipids or fatty acids are used for compartmentalization[5–8]. However, in principle also other suitable amphiphilic building blocks can serve as membrane material, such as peptides or synthetic block copolymers. For instance, Huber et al.[9] used a specifically designed amphiphilic elastin-like peptide (ELP) to form vesicles inside E. coli. The genetic template for the production of such peptides can be enclosed inside an in vitro reaction compartment and therefore directly linked to the expression machinery. In order to implement this in a cell-free context[10,11], transcription-translation systems based on purified components, e.g., the PURE system[7,12], or on bacterial cell extracts, e.g., the TX-TL system[13,14], can be used. These systems employ the multicomponent bacterial translation machinery, to express proteins from externally added DNA in a one pot reaction. For the transcription the T7 polymerase can be used as well as the present constitutive based transcription system in the case of the E. coli TX-TL system.

Such in vitro systems have been successfully used to implement and to study gene expression, gene circuits, expression and DNA self-replication[12,15–18]. For instance, cell-free systems were encapsulated into phospholipid vesicles to facilitate compartmentalized protein expression for several hours[15].

In the present work, we encapsulate the TX-TL system in peptidosomes made of amphiphilic elastin-like peptides (ELP). These peptides can be easily expressed in cell-free systems[10,11] and thus they simplify the synthesis of the membrane material in comparison to lipid synthesis. We show that biomolecules can be easily enclosed in ELP-based vesicles, and we further demonstrate in vesiculo transcription of an RNA aptamer and translation of a fluorescent protein. We finally, succeed in the expression of the membrane-constituting peptides inside the vesicles themselves and demonstrate their incorporation into the membrane and thus inherent vesicle growth.

## Results

**The synthetic cell model**. The molecular building block of our membrane was a synthetic peptide derived from the protein tropoelastin, which commonly comprises the sequence motif $(G\alpha GVP)_n$, where $\alpha$ can be any natural amino acid except for proline and $n$ is the number of pentapeptide repeats. ELPs are stimulus-sensitive peptides and undergo a fully reversible phase transition from a hydrophilic to a hydrophobic state when the sample temperature exceeds the specific transition temperature $T_t$[19]. The latter depends on several parameters such as the amino acid used for $\alpha$, concentration, salt conditions, pH, etc. Unlike other hydrophobic molecules such as lipids, ELPs in a hydrophobic coacervate state can still exhibit a water content of about 63% by weight[20].

In order to create an amphiphilic peptide capable of membrane formation, we produced a diblock copolymer with the sequence MGH-GVGVP((GEGVP)$_4$(GVGVP))$_4$—((GFGVP)$_4$(GVGVP))$_3$(GFGVP)$_4$-GWP abbreviated as EF. At physiological pH, this peptide has a charged hydrophilic E-rich block (mainly GEGVP pentapeptides) with a specific $T_{t,E}$ below sample temperature $T$ and a hydrophobic F-rich block (mainly GFGVP pentapeptides) with a specific $T_{t,F}$ above $T$ (Fig. 1a). The peptide was expressed using E. coli cells carrying a plasmid coding for EF and purified using inverse transition cycling[21] at pH 2 and 7 (see Methods). The purity of the protein was confirmed by sodium dodecyl sulfate–polyacrylamide gel electrophoresis (SDS-PAGE) with Roti®-blue staining (Supplementary Fig. 1) and the concentration was determined using spectroscopic methods (see Methods). For controlled formation of vesicles, the peptides were dissolved in a chloroform-methanol mixture together with spherical glass beads (see Methods). We assume that the vesicle formation process is similar to the liposome formation process (Fig. 1b)[7]. After fast evaporation of the organic solvent the glass beads are coated with EF. Due to the addition of the swelling solution (initially 1x phosphate-buffered saline (PBS)) the EF film rehydrates through budding and the swelling solution is encapsulated. Using dynamic light scattering (DLS) we determined the diameter distributions for 110, 180, 220, and 440 pM EF. The corresponding peak values were 87, 178, 220, and 250 nm with dispersion (sample standard deviation) values of 47, 67, 111, and 415 nm (Supplementary Table 1). For further experiments we used 180 pM EF since it resulted in the lowest relative dispersion (dispersion divided by the peak value). Using transmission electron microscopy (TEM) we determined the peak value of the diameter distribution for 180 pM EF (Fig. 1c) to 176 nm with a dispersion of 68 nm, which is in good agreement with the DLS data (Supplementary Fig. 2). The size distribution of the TEM and DLS data can be described by a Weibull extremal probability distribution[22] (Supplementary Note 1), and all given dispersion values are determined from a Weibull fit to the data. Membrane formation and its stability depend on the osmotic pressure, the critical aggregation concentration and the chemical potentials in general. Therefore, the samples were not diluted if applicable or purified.

We also verified the size stability of the vesicles over 18 h using DLS with a mean of 179 nm and a standard deviation of 10 nm (Supplementary Fig. 3). We found consistent mean diameters in DLS measurements from five repetitive rehydration experiments indicating no stochastic influence on the vesicle formation process with a student-$t$ corrected standard deviation of 12.4 nm for a $P$-value of 0.95. For a swelling solution containing 1x PBS, the mean membrane thickness was roughly determined to be 4.9 nm with a standard deviation of 0.5 nm using TEM (Supplementary Table 2 and Supplementary Fig. 8), which is in good agreement with Huber et al. who used an amphiphilic ELP of similar size in a GFP-ELP fusion protein. Furthermore, we demonstrated vesicle stability at various NaCl membrane gradients (Supplementary Fig. 5 and 6). In order to show encapsulation using the glass beads method, we utilized fluorescently-labeled DNA in the swelling solution. The resulting vesicle sample was diluted and subsequently measured with a flow cytometer. The fluorescence intensities of the vesicles scaled with the DNA concentrations used (Fig. 1d and Supplementary Fig. 7). In a control experiment without vesicles the fluorescence determined for a sample of FAM-labeled DNA was reduced due to dilution of the DNA, but it also scaled with the concentration (Supplementary Fig. 20).

**Transcription in vesicles**. We next studied the efficacy of in vitro transcription reactions encapsulated inside of the peptide vesicles. To this end, we transcribed the fluorogenic dBroccoli RNA aptamer[23] in the presence of its cognate fluorophore DFHBI (3,5-difluoro-4-hydroxybenzylidene imidazolinone) and monitored

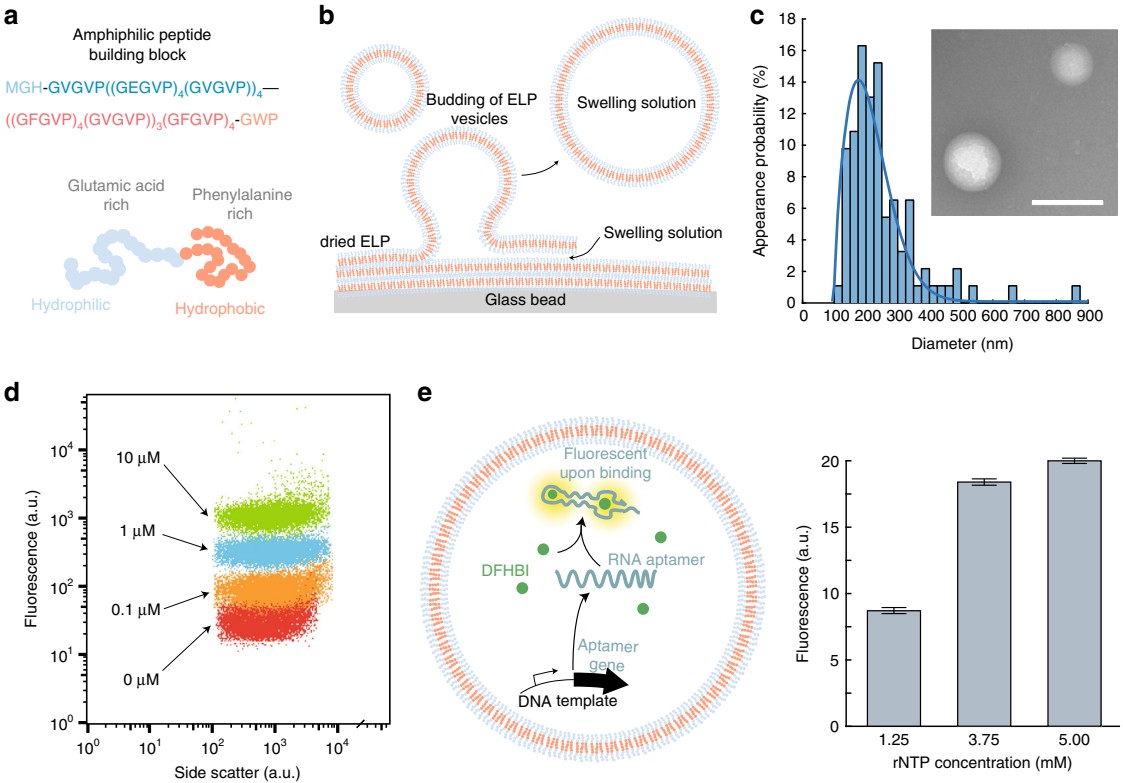

**Fig. 1** ELP vesicle formation and transcription. **a** The peptide building block is an amphiphilic ELP with a hydrophilic glutamic acid-rich domain and a hydrophobic phenylalanine-rich domain. **b** For vesicle formation dried ELPs are rehydrated from glass beads. **c** Size distribution of the produced vesicles obtained from TEM measurements with diameter of 176 nm. The data are described using a Weibull probability distribution (solid line). The inset shows a typical TEM image. Scale bar: 200 nm. **d** Co-localization of fluorescently labeled DNA (with indicated concentrations) into the vesicles measured by flow cytometry. **e** Left: Illustration of the transcription of an RNA aptamer inside of an ELP vesicle. The aptamer binds to DFHBI, which then fluoresces. Right: Plateau value means of the DFHBI fluorescence after 50 min of transcription for various rNTP concentrations. The given error bars indicate the sample standard deviation of the measured plateau values

the increasing fluorescence of the vesicles over time. dBroccoli is the dimeric version of the Broccoli aptamer, which exhibits robust folding, even in low magnesium concentrations, and it's optimized for usage in living cells. DFHBI is a small non-fluorescent molecule that gets into a highly fluorescent state upon binding to its aptamers such as Broccoli. The encapsulated transcription mix (TX) consisted of ribonucleoside tri-phosphates (rNTPs), electrolytes, DFHBI (dimethyl sulfoxide (DMSO) stock), T7 RNA polymerase, and the DNA template for the dBroccoli aptamer. Encapsulation was performed with the glass beads method described above, whereby 10% DMSO in solution did not affect vesicle formation. To suppress transcription outside of the vesicles DNase I was added to the outer solution after formation of the vesicles, which digested any non-encapsulated DNA template. As a negative control, the DNase I was added before the vesicle formation was done.

As expected, transcription inside the vesicles led to an increase in the fluorescence signal, which reached its maximum after ≈50 min. The plateau phase was most probably caused by the depletion of resources such as the rNTPs, the formation of pyrophosphates or the exhaustion of the polymerase. To test the hypothesis of resource depletion the rNTP concentration was altered. The measured intensity values always reached their plateau phase after the same time but depending on the rNTP concentration the maximum fluorescence level increased (Fig. 1e and Supplementary Fig. 9). This increase was not linear with the rNTP concentration, and hence the rNTP depletion was not solely responsible for the limitation of the transcription reaction.

**Protein expression in peptide vesicles**. In order to demonstrate a compartmentalized transcription–translation process, we expressed the fluorescent protein mVenus (Fig. 2a) and YPet (Supplemental Figs. 10 and 11) inside of the vesicles. Encapsulation again was carried out using the glass beads method. Upon gene expression, mVenus fluorescence increased and reached a plateau phase after 180 min (Fig. 2 and Supplementary Fig. 10). Expression of proteins occurring outside of the vesicles was suppressed using the antibiotic kanamycin, which blocks the 30S-subunit of the ribosome and thus prevents translation. As an alternative, we used EDTA to chelate magnesium ions in the buffer and thereby also suppress gene expression (Supplementary Fig. 11). Neither the presence of kanamycin nor EDTA outside of the vesicles compromised expression of mVenus inside of the peptidosomes, which indicates that the peptide membrane is not permeable for these small molecules on the time-scale of our experiments. When kanamycin or EDTA was encapsulated together with the expression mix as a control, protein expression was successfully suppressed (Fig. 2b red curve and Supplementary Figs. 10 and 11). The TX-TL system also contains cofactors, such as $NAD^+$ or FAD, which change their autofluorescence upon reduction to NADH or $FADH_2$ and thus create an additional change in the fluorescence signal. Therefore, we used a background correction with a sample containing the cell extract, but without plasmid. By assuming a Poisson distribution we can calculate the probability to find the TX-TL components inside the vesicles (Supplementary Note 2). For concentrations of about 1 μM and above (e.g., the proteins)

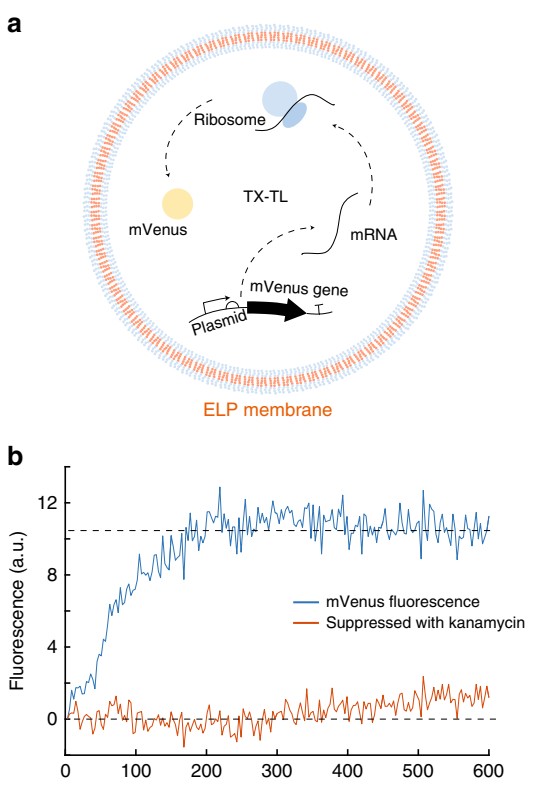

**Fig. 2** Protein expression inside ELP vesicles. **a** Illustration of the expression of mVenus inside an ELP vesicle containing bacterial cell extract TX-TL. **b** Time-dependent fluorescence of expressed mVenus (blue) and fluorescence of kanamycin-suppressed expression (red). The dashed lines are a guide to the eye

the probability is close to 100%, whereas for the plasmid (50 nM) the probability is 35%.

**Vesicle growth caused by compartmentalized peptide synthesis.** As a proof of principle, we measured the growth of the peptide vesicles, when monomers are added to the outer solution. In DLS measurements we could measure a size change from 192 nm (dispersion 74 nm) to 234 nm (dispersion 101 nm) after the addition of 50 µM ELP (Supplementary Fig. 4).

Given the capacity for protein expression within the peptidosomes, we finally proceeded to synthesize the membrane-constituting peptide itself inside of the vesicles (Fig. 3a). To verify peptide expression inside the vesicles containing the TX-TL system, we equipped EF with a His-tag. After vesicle formation and EF expression we analyzed this sample for His-tagged peptides. External expression was suppressed using kanamycin. Using western blotting the peptide was clearly identified after 240 min of expression, whereas in the initial sample (before expression) no His-tagged peptide was found (Fig. 3b bottom and Supplementary Fig. 18). The peptide band was shifted upward with respect to the expected position at about 18 kDa, which is a well-known effect for elastin-like polypeptides[24,25]. For clarification we confirmed the expected weight of our non-tagged EF construct by mass spectrometry. We found a peptide mass of 18180 Da (Fig. 3b top and Supplementary Fig. 21), which is the exact ELP mass reduced by the mass of the methionine at the beginning of the peptide sequence, which most probably has been removed through posttranslational modification[26]. The two peaks

next to the parent mass most likely represent the peptide mass with $Na^+$ and the peptide mass with acetonitrile.

In the next step, we investigated the incorporation of the internally generated EF peptides into the vesicle bilayer and thus the growth of the membrane from within. The vesicles were produced as before and kanamycin was utilized to suppress outside expression. After 8 h of incubation, flow cytometry measurements indeed showed an increase of the forward scattering signal. This indicates a growth in vesicle size with respect to a reference sample where kanamycin was present inside the vesicles (Supplementary Fig. 19).

Using TEM we statistically verified the relative size change of the vesicles (Supplementary Figs. 12 and 13). The freshly prepared vesicles, which were able to grow, were divided into two batches. One was immediately flash-frozen to suppress peptide expression, whereas the second was incubated at 29 °C for 240 min and then flash-frozen to stop expression. At the beginning of the expression (before incubation of the sample) the peak of the size distribution was found at a diameter of 157 nm with a dispersion of 104 nm, while peptide synthesis for 240 min resulted in a diameter of 330 nm and a dispersion of 83 nm (Fig. 3c, Supplementary Fig. 16, and Supplementary Table 3). As mentioned before only 35% of the vesicles should contain a plasmid, and are therefore able to express EF and to grow. Figure 3c shows no indication of a not growing subpopulation at $t = 240$ min. We assume that the vesicles exchange membrane peptides, which makes the whole population grow; perhaps this also indicates the existence of flip-flop between the leaflets. As a negative control, the hydrophilic ELP $(GVGVP)_{40}$ (further denoted as V40) was expressed to keep the load on the expression system similar. Neither flow cytometry nor TEM measurements (Supplementary Figs. 14 and 15) showed a measurable change in vesicle size in this case, which is in agreement with the fact that V40 is not able to incorporate into the membrane. The peak value determined from these TEM measurements was 149 nm with a dispersion of 102 nm at the beginning of V40 expression and 145 nm with a dispersion of 128 nm after 240 min.

To further examine vesicle growth, we utilized a Förster resonance energy transfer (FRET) assay to monitor the incorporation of internally expressed EF into the membrane. To this end, we prepared two batches of EF, which were either modified with the fluorophore Cy3 or the fluorophore Cy5 via copper catalyzed azide-alkyne Huisgen cycloaddition (see Methods). Vesicles were then formed using a 1:1 mixture of Cy5-EF and Cy3-EF. As a result, the dyes were randomly and homogeneously distributed within the vesicle membrane after the formation with the glass beads method. As Cy3 and Cy5 constitute a FRET pair, excitation of the Cy3 fluorophore, therefore, led to fluorescence emission of the Cy5 acceptor via FRET.

In a bulk experiment, we found that expression of non-labeled EF within the vesicles led to a decrease in the acceptor signal accompanied by an increase in donor fluorescence. This indicated an increasing average distance between the dyes inside the membrane (Fig. 3d, e) and thus an incorporation of new EF. The vesicles were produced as before and kanamycin was utilized to suppress outside expression. In control measurements with kanamycin inside the vesicles, the FRET signal stayed constant (Supplementary Fig. 17). This clearly demonstrates that EF peptides expressed in the interior of peptide vesicles incorporate into the membrane and cause vesicle growth.

From the measured vesicle diameters, we could estimate the relative number of peptides expressed inside the vesicles. The membrane volume at times 0 min and 240 min after start of the expression reaction can be calculated using the volume of a single ELP $V_{ELP}$ and the number of peptides $N_0$ and $N_{240}$ at these times.

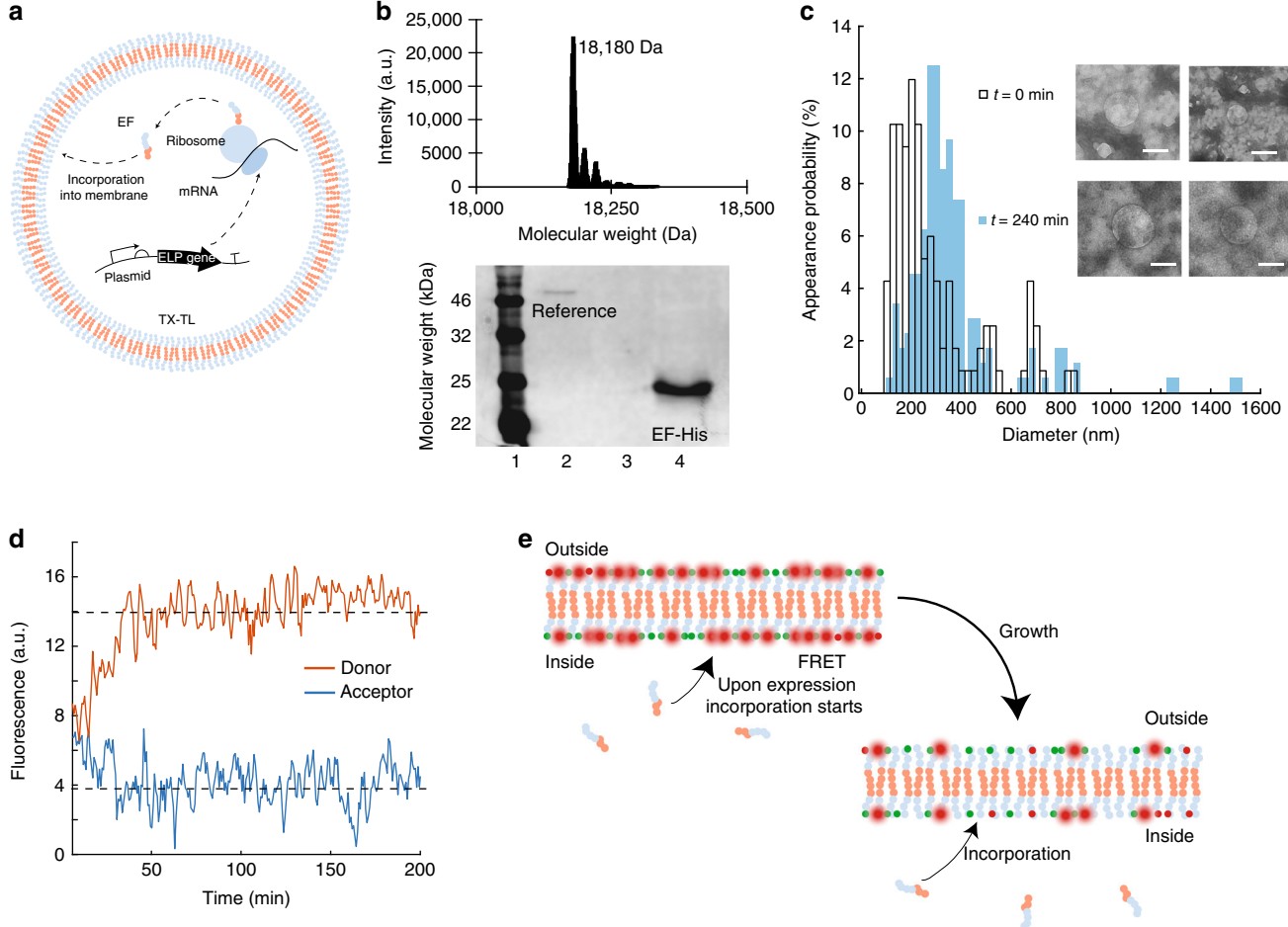

**Fig. 3** Expression of ELP and vesicle growth. **a** Illustration of the expression of EF inside a vesicle using TX-TL. **b** Top: Mass spectroscopy of full length EF. Bottom: Western blot of EF-His expressed in TX-TL within vesicles after 0 min (lane 3), where no EF-His is detected and after 240 min (lane 4). The reference is a histidine-tagged helicase (lane 2). **c** Size distribution of vesicles at the beginning of EF expression ($t = 0$ min) with a peak value of 149 nm and after 240 min with a peak value of 330 nm using TEM. Inset: Typical TEM images at $t = 0$ min (top) and $t = 240$ min (bottom). Scale bar: 200 nm. **d** Donor fluorescence (red) of dye-labeled EF located in the membrane and acceptor fluorescence (blue) of a labeled EF. The dashed lines are a guide to the eye. **e** Illustration of the FRET assay used. The vesicles are formed using Cy5-EF and Cy3-EF. Upon expression of unlabeled EF and its incorporation into the membrane the mean distance between the FRET pairs rises and the donor signal increases

The relative volume increase of the membrane $\xi = N_{240}V_{ELP}/N_0V_{ELP} = N_{240}/N_0$ is related to the vesicle radii at times 0 min and 240 min via the expression $\xi \approx R^2_{240}/R^2_0$ (Supplementary Note 3). From our experiments, we found $\xi \approx 4.4$, which means a 4.4-fold increase of the initially present number of ELP due to peptide production inside the vesicles.

## Discussion

In conclusion, our results demonstrate that peptide vesicles are promising candidates for the generation of artificial cell-like compartments. Their fabrication is relatively straightforward and the encapsulation of biochemical reaction mixtures is—at the moment—only limited for low-concentrated molecules by the vesicle size. We successfully showed transcription of an RNA aptamer and the expression of fluorescent proteins inside our peptide vesicles. Most importantly, we demonstrated vesicle growth through expression of the membrane peptide in vesiculo and its incorporation into the membrane.

It is conceivable that in future work peptide vesicle growth and also replication of the encapsulated genetic templates could be coupled, which would be a major step towards the generation of self-replicating protocellular compartments.

## Methods

**Expression of elastin-like peptides**. For bacterial ELP expression, the peptide gene was cloned into a *pET20b(+)* expression vector (Novagen) and transformed into the *BL21(DE3)pLysS* strain of *E. coli*. As confirmed by Sanger sequencing, the gene encoded the polypeptide sequence MGHGVGVP((GEGVP)$_4$(GVGVP))$_4$ ((GFGVP)$_4$(GVGVP))$_3$(GFGVP)$_4$GWP (abbreviated as EF). ELP expression was performed in a culture flask shaker at 37 °C in a 1 L culture of LB (Luria/Miller) medium (10 g of tryptone, 5 g of yeast extract, 10 g of NaCl, and 100 mg of carbenicillin), induced with 240 mg of IPTG (isopropyl β-D-1-thiogalactopyranoside) when the optical density at 600 nm reached approximately 0.8. After 16 h of incubation at 16 °C, the bacteria were harvested through centrifugation. The bacteria were lysed by sonication in phosphate-buffered saline (PBS, pH 7.4) supplemented with lysozyme (1 mg/mL), 1 mM pMSF, 1 mM benzamidin and 0.5 U of DNase I. After lysis 2 mL of 10% (w/v) PEI was added per 1 L of original cell culture. The samples where incubated at 60 °C for 10 min and afterwards at 4 °C for 10 min, followed by a centrifugation at 16,000 x *g* at 4 °C for 10 min.

The ELPs were purified through sequential centrifugations under acidic (pH 2) and neutral (pH 7) conditions. For the pH adjustment phosphoric acid and sodium hydroxide were used. During centrifugations at pH 7, the ELPs remained in the supernatant, while during centrifugations at pH 2 they phase-separated into the pellet, which was re-suspended in water. After three cycles of centrifugation, the ELPs were dissolved in water at a concentration of 700 µM. The concentration of the peptides was measured using absorption spectrometry (Nanophotometer

IMPLEN vers. 7122 V2.3.1, Munich, Germany), assuming an extinction coefficient of 5500 M/cm at 280 nm.

**Glass beads method**. Two-hundred microliters of concentrated 1.1 mM ELP solution was mixed with 1250 μL of a 2:1 chloroform/methanol mixture, for fast evaporation. A total of 1.5 g of spherical glass beads (212 μm to 300 μm in size) were added to a round-bottom flask. Using a rotary evaporator the solvent was evaporated, resulting in a peptide film on the glass beads. For further experiments 100 mg of the glass beads were mixed with 60 μL of the swelling solution containing the molecules to be encapsulated. After an incubation for 5 min at 25 °C, the vesicles had formed and the sample was centrifuged to sediment the glass beads. The vesicle solution was removed using a pipette.

**Western blotting**. Samples were mixed with 2x Laemmli buffer and heated to denature the peptide structure. SDS PAGE (12%) was used for separation of the sample components. For further analysis, the peptides were fixed on a PVDF (polyvinylidene difluoride) membrane by transferring the content of the SDS gel to the membrane using a Semi-Dry blotter. The peptide-free areas of the membrane were blocked by incubation in a blocking solution containing bovine serum albumine (BSA). Afterwards, the membrane was rinsed several times with PBST (phosphate buffered saline with Tween 20) to remove residual BSA. The detection of the immobilized peptides was carried out by incubating the membrane with a specific anti-His antibody (6 × -His Epitope tag antibody, mouse, purchased from Life Technologies GmbH, Darmstadt, Germany: catalog number MA1135, clone 4E3D10H2/E3) at 4 °C overnight. Residual antibodies were removed by washing with PBST. For visualization secondary antibodies (anti-mouse Alexa Fluor 680, goat, purchased from Life Technologies: catalog number A28183) were added onto the membrane and incubated for 1 h at room temperature. Residual secondary antibodies were removed through washing with PBST before the membrane was imaged a fluorescent scanner (Typhoon Fla 9500, GE Healthcare Life Science) (Fig. 3b bottom and Supplementary Fig. 18 for uncropped scan). Both antibodies were used at dilutions of 1:1000.

**Transcription reaction (TX)**. The sequence of the RNA aptamer dBroccoli was GAGACGGTCGGGTCCATCTGAGACGGTCGGGTCCAGATATTCGTATCT GTCGAGTAGAGTGTGGGCTCAGATGTCGAGTAGAGTGTGGGCTC[23]. The transcription solution contained 1x RNAPol reaction buffer (40 mM Tris-HCl (pH 7.9), 6 mM MgCl$_2$, 1 mM dithiothreitol (DTT), 2 mM spermidine), 125 mM KCl, 15 mM MgCl$_2$, 4 mM rNTP, 10 μM DFHBI, 200 nM DNA template, 4 U/μL T7 polymerase (NEB, M0251S), 0.5 U/μL RNase inhibitor murine (NEB, M0314S) and water. All measurements took place at 37 °C.

**Transcription translation reaction (TX-TL)**. For the generation of crude S30 cell extract a *BL21-Rosetta 2(DE3)* mid-log phase culture was bead-beaten with 0.1 mm glass beads in a Minilys homogenizer (Peqlab, Germany) as described in ref. [27]. The composite buffer contained 50 mM Hepes (pH 8), 1.5 mM ATP and GTP, 0.9 mM CTP and UTP, 0.2 mg/mL tRNA, 26 mM coenzyme A, 0.33 mM NAD, 0.75 mM cAMP, 68 mM folinic acid, 1 mM spermidine, 30 mM PEP, 1 mM DTT and 2% PEG-8000. As an energy source in this buffer phosphoenolpyruvate (PEP) was utilized instead of 3-phosphoglyceric acid (3-PGA). All components were stored at −80 °C before usage. A single cell-free reaction consisted of 42% (v/v) composite buffer, 25% (v/v) DNA plus additives and 33% (v/v) S30 cell extract. For ATP regeneration 13.3 mM maltose and 1 U of T7 RNA polymerase (NEB, M0251S) were added to the reaction mix[2]. All measurements took place at 29 °C with 50 nM of plasmid if it is not indicated differently.

**Dynamic light scattering**. For the DLS experiments the instrument DynaPro Nanostar (Wyatt technology corporation) was used. The buffers were sterile filtered before usage and the samples were measured in a disposable cuvette. For one distribution a set of 50 single measurements where performed for 2 s and averaged afterwards. The values were averaged and processed with the DYNAMICS software using a CONTIN-like algorithm.

**Transmission electron microscopy**. The vesicle solution was adsorbed on glow-discharged formvar-supported carbon-coated Cu400 TEM grids (FCF400-CU, Science Services, Munich, Germany) for 2 min, followed by a negative stain using a 2% aqueous uranyl formate solution with 25 mM sodium hydroxide for 45 s. Afterwards the grid was dried and stored under vacuum for 30 min. Imaging was carried out using a Philips CM100 transmission electron microscope at 100 kV. For acquiring images an AMT 4 megapixel CCD camera was used and imaging was performed at magnification between x 8900 and × 15,500. For image processing the plugin Scale Bar Tools for Microscopes for Java-based software ImageJ was used.

**Flow cytometry**. The flow cytometer measurements were performed by using a CyFlow Cube 8 cytometer (Sysmex Partec GmbH, Germany) equipped with a blue laser emitting at 488 nm. The measured signals were the forward scattering signal (FSC), the side scattering signal (SSC) and the fluorescence signal, which was band-pass filtered at 536 nm ± 40 nm. The buffers were sterile filtered and degassed before usage. For a measurement 100 μL of the sample was diluted with 500 μL 1 x PBS (8 g/L NaCl, 2 g/L KCl, 1.42 g/ L Na$_2$HPO$_4$, 0.27 g/L K$_2$HPO$_4$, pH 6.8–7.0) and measured immediately. The analysis of the data was performed with the FlowJo v10 software (FlowJo LLC, USA).

**Fluorescence measurements**. Cell-free expression and transcription was characterized via plate reader measurements, with the corresponding filter sets for the fluorescence (BMG FLUOstar Optima) using 15 mL reaction volumes in 384-well plates.

**FRET measurements**. The vesicles were prepared according to the glass beads method, with a mixture of 100 μL of 1.1 mM Cy3 labeled ELPs and 100 μL of 1.1 mM Cy5 labeled ELPs. For rehydration the TX-TL solution was used with the plasmid containing the EF gene (50 nM). Expression of EF occurring outside of the vesicles was suppressed using the antibiotic kanamycin. For the reference, kanamycin was added before the vesicle formation. Cell-free expression was characterized via plate reader measurements in a bulk measurement, with the corresponding filter sets for the FRET dyes (BMG FLUOstar Optima) using 15 mL reaction volumes in 384-well plates.

**Mass spectrometry**. Full-length protein mass spectrometry was performed on a Dionex Ultimate 3000 HPLC system coupled to a Thermo LTQ-FT Ultra mass spectrometer with electrospray ionization source (spray voltage 4.2 kV, tube lens 120 V, capillary voltage 48 V, sheath gas 60 arb, aux gas 10 arb, sweep gas off). In all, 2.5 μL of sample corresponding to 1.64 nmol of peptide were on-line separated using a BioBasic-4 column (Thermo; 150 mm × 1 mm, 5 μm) by applying a multistep gradient from 2% to 20% eluent B over 6 min; 20% to 25% B over 1 min and 25% to 85% B over 14 min (eluent A: water with 0.1% (v/v) formic acid; eluent B: 90% (v/v) water, 10% (v/v) acetonitrile with 0.1% (v/v) formic acid; flow: 0.2 mL/min). All solvents were of liquid chromatography-mass spectrometry grade. The mass spectrometer was operated in positive mode collecting full scans at $R = 50,000$ from $m/z$ 400 to $m/z$ 2000. Collected data was deconvoluted using Thermo Xcalibur Xtract algorithm.

**Click chemistry**. Initially, elastin-like peptides were activated with an azide group and then conjugated to dyes via copper-based azide-alkyne Huisgen cycloaddition (denoted as click chemistry). The used NHS-azide linker (γ-azidobutyric acid oxysuccinimide ester) was diluted in DMSO to a final concentration of 20 mM. ELPs were dissolved in 1x PBS (8 g/L NaCl, 2 g/L KCl, 1.42 g/ L Na$_2$HPO$_4$, 0.27 g/L K$_2$HPO$_4$, pH 6.8–7.0). The peptides were mixed with a 2-fold excess of NHS-azide and incubated for 12 h at room temperature. To remove residual NHS-azide, the sample was loaded into a 10 kDa dialysis cassette and stored at 4 °C for 12 h. In the next step, the activated EF and dye were conjugated using the aforementioned click chemistry. The alkyne-modified dye was mixed with activated EF at a molar ratio of 1:1 (dye:ELP). Afterwards 1 mM TBTA (tris(benzyltriazolylmethyl)amine), 10 mM TCEP (tris(2-carboxyethyl)-phosphine hydrochloride), and 10 mM CuSO$_4$ were added; all given concentrations are final concentrations. The mixture was incubated at 4 °C for 12 h. Remaining linker strands were removed by dialysis with a 10 kDa dialysis cassette, which was stored at 4 °C for 12 h.

## Data availability

The authors declare that the main data supporting the findings of this study are available within the article and its Supplementary Information file. Extra data are available from the corresponding author upon request.

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

## Acknowledgements

We gratefully acknowledge financial support through the European Research Council (grant agreement no. 694410 – AEDNA), the TUM International Graduate School for Science and Engineering IGSSE project no. 9.05 (to M.A.G and T.P.), and the Cluster of Excellence Nanosystems Munich (NIM). We thank N.B. Holland for discussion and E. Falgenhauer for her help with sample preparation. We thank A. Dupin and M. Schwarz-Schilling for their help with the TX-TL system and useful discussions.

## Author contributions

K.V. and T.P. designed research. K.V. performed research, T.F., L.G. and M.A.G. assisted in experiments. K.V., F.C.S., and T.P. analyzed data. M.H. and S.A.S. performed and analyzed the mass spectrometry measurements, K.V., F.C.S., and T.P. wrote the manuscript.

## Additional information

**Competing interests:** The authors declare no competing interests.

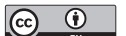

