## [Peer Review File · Nature Communications]

Reviewers' Comments:

Reviewer #1:

Remarks to the Author:

This paper describes construction of semi-synthetic minimal cells containing cell-free protein expression system encapsulated inside protein vesicles.

The work described in this paper is very well designed, the text is written clearly and coherently and project description is very logical.

All figures show TxTI expression only INSIDE the vesicle. I don't see purification steps mentioned at any point in the text or in methods. The vesicle formation method, as described, would leave TxTI both inside and outside of the vesicles. But on Figure 3e the membrane has inside and outside sides labeled.

Please clarify how was the unencapsulated, external TxTI inactivated or removed.

Can the vesicle forming peptide perform flip-flop in the membrane? The images of spherical growth of the vesicles would suggest the peptide synthesized inside vesicles (assuming it is not being synthesized outside as well - see my comment above) is able to equilibrate between inner and outer leaflet, to allow for symmetrical membrane growth.

What kind of membrane concentration gradient those peptide membranes can withstand? Again, spherical image of "after growth" suggests the osmolarity is not an issue, at least at the observed growth rate.

What is the encapsulation efficiency of the vesicles? In other words, what percentage of peptide liposomes contains all elements of the complete transcription and translation system?

In the broccoli transcription experiments, how was DFHBI added?
if it was DMSO stock, it would be important to see if the final DMSO concentration affect stability of the peptide membrane.

Was inorganic pyrophosphatase used for the transcription reaction? Or was the transcription yield low enough that there would be no problem with pyrophosphates precipitating with magnesium?

The size of the vesicles is smaller than typical sizes used in TxTI reactions, most publications use liposomes around 500nm and larger.

Was the small size designed on purpose, or it's just inherent property of glass bead vesicle formation?

The author mention that they used phosphoenolpyruvate (PEP) instead of 3-phosphoglyceric acid (3PGA) as the energy source. Was the reason for that simple the world-wide shortage of 3PGA we're currently experiencing? or was there another reason for using PEP instead of 3PGA? It would be particularly useful to know if authors noticed any advantages of one energy system over the other in the peptide vesicles, making it differ from TxTI performance in lipid vesicles

I don't like "guide to the eye" lines on figure 2. There is nothing technically wrong with it, and authors clearly stated those are not fits. Please treat this more as a comment than suggestion: in my opinion, lines on charts should only come from data and fits.

I think this is very valuable and interesting paper, and it would be of great interest to the broad readership of Nature Communications.

Reviewer #2:

Remarks to the Author:

The manuscript "Towards synthetic cells using peptide-based reaction compartments" by Vogele et al. describes the generation of peptide vesicles as potential artificial cell-like compartments. Successful transcription (RNA aptamer) and translation (fluorescent protein) inside the peptide vesicles is demonstrated. This was followed by expression of the vesicle peptide itself and evidence of incorporation of the peptide into the vesicle membrane which enabled vesicle growth from within.

This work is a major step towards the development of artificial cell compartments based on non-lipid membranes and towards the generation of protocells. The manuscript is well written but should be expanded, particularly in the introduction and discussion to give more depth to the work. Other potential applications for such peptide vesicles e.g. drug delivery should be explored. The figures are accurately presented and the claims made are novel and will be of great interest to many others in the field.

Comments:

Clearer description and more detailed graphical representation of the peptide building blocks and proposed vesicle formation is required. From Figure 1a and Figure 1b it is difficult to see how the peptide is thought to assemble. It would be useful to describe the rationale for the design of the EF peptide.

The authors demonstrate that the ELP vesicles are formed in vitro and provide electron microscopy images of vesicles to evidence their claim. When swelling the peptides, the vesicles reach round 180 nm in diameter. Is anything known about the size-determining factors? Is the size peptide concentration-dependent and can it be modified by changing the peptide concentration or other conditions? Have experiments carried out to test various conditions? If so, the findings should be included in the manuscript.

ELP vesicles have been shown to assemble in vivo by Huber et. al. Can ELP vesicles self-assemble in vitro without prior adsorption onto glass beads?

The authors show encapsulation of fluorescent DNA by flow cytometry as an indirect measure of encapsulation. It would be more convincing to show light microscopy images of vesicles containing the fluorescent dye. I suggest such images are added.

When encapsulating cell-free transcription/translation the vesicle samples contain many impurities (seen in Figures S8-S11) and vesicles are not clearly visible for size determination. Transcription/translation outside the vesicles needed to be inhibited. Have the authors attempted to purify the "loaded" vesicles to avoid some of these issues?

Membrane thickness measurements are presented. If I understand correctly these measurements have been carried out on intact vesicles deposited on EM grids. However, the membrane thickness cannot be measured without sectioning through the vesicles. If the authors want to compare membrane thickness, the vesicles should be embedded in resin and sectioned through.

Other comments:

Introduction:

Briefly explain the TX-TL system.

"These peptides can be easily expressed in cell-free systems and thus they simplify the synthesis"
Have these peptides been expressed in cell-free before?

Results:

It is mentioned that the ELP becomes hydrophilic or hydrophobic depending on the incubation temperature. This needs to be better explained.

The EF protein has been purified and purity confirmed by SDS gel and mass spectrometry. The mass spec data and gel should be shown in the supplementary.

The dBroccoli RNA aptamer needs to be better explained.

Figure 3d: The reference protein (His-tagged helicase) is hardly visible on the gel.

Methods:

Western blotting:

"specific anti-His antibody" Which antibody was used?

Supplementary:

Vesicle size distribution: S8, S9, S10, S11. In most of these images it is very difficult to see the vesicles and it is unclear what all the different images are. Are they multiple examples of the same sample?

S12 shows "size distribution from about 100 data points" Are you referring to the measurement of 100 vesicles?

Dear Reviewers,

we are very thankful for reviewing our manuscript. We revised our manuscript (changed text is highlighted in Word file) and the Supplementary Information (list of changes also submitted). Please find attached our point-by-point replies to the reviewers' comments.

Tobias Pirzer

Reviewers' comments:

Reviewer #1 (Remarks to the Author):

This paper describes construction of semi-synthetic minimal cells containing cell-free protein expression system encapsulated inside protein vesicles.

The work described in this paper is very well designed, the text is written clearly and coherently and project description is very logical.

All figures show TxTI expression only INSIDE the vesicle. I don't see purification steps mentioned at any point in the text or in methods. The vesicle formation method, as described, would leave TxTI both inside and outside of the vesicles. But on Figure 3e the membrane has inside and outside sides labeled.

Please clarify how was the unencapsulated, external TxTI inactivated or removed.

-> This is correct, TX-TL was always present at the outside. To suppress peptide expression at the outside we added kanamycin to the outer solution after vesicle formation. We added a sentence to the manuscript where it was appropriate.

Can the vesicle forming peptide perform flip-flop in the membrane? The images of spherical growth of the vesicles would suggest the peptide synthesized inside vesicles (assuming it is not being synthesized outside as well - see my comment above) is able to equilibrate between inner and outer leaflet, to allow for symmetrical membrane growth.

-> This is an excellent question. Since the vesicles are always of spherical shape without any noticeable deformations, eversion or even buds, we assume that the peptides can perform flip-flop in the membrane. Up to now, we don't have a direct proof of this, but this will be studied in the future.

What kind of membrane concentration gradient those peptide membranes can withstand? Again, spherical image of "after growth" suggests the osmolarity is not an issue, at least at the observed growth rate.

-> Thank you for this question. We tested the stability of the membranes for four salt gradients by adding NaCl to the outer solution. In DLS and TEM measurements we see that size and integrity are not affected. Only at an outside NaCl concentration of 1 M the vesicles become slightly ellipsoid. We added these data into the SI.

What is the encapsulation efficiency of the vesicles? In other words, what percentage of peptide liposomes contains all elements of the complete transcription and translation system?

-> This is indeed an important point. If we assume a Poisson distribution to derive the encapsulation probability we find encapsulation probabilities of about 100% for all components for low μM concentrations or higher. The plasmid was the only element below that concentration and with only 50 nM it has an encapsulation probability of 35%. We added this estimation to the main text and in detail to the SI.

In the broccoli transcription experiments, how was DFHBI added?

if it was DMSO stock, it would be important to see if the final DMSO concentration affect stability of the peptide membrane.

-> This is correct. The used DFHBI was added from a DMSO stock. We performed an additional experiment to show that the membrane stability is not affected. 10% DMS in 1x PBS does not affect the vesicle formation or stability. We added this information to the main part.

Was inorganic pyrophosphatase used for the transcription reaction? Or was the transcription yield low enough that there would be no problem with pyrophosphates precipitating with magnesium?

-> We didn't use inorganic pyrophosphatase. We think that our transcription yield is mainly influenced by the depletion of rNTPs. But as mentioned in the text we cannot entirely dismiss complexation of magnesium by pyrophosphates.

The size of the vesicles is smaller than typical sizes used in TxTI reactions, most publications use liposomes around 500nm and larger.

Was the small size designed on purpose, or it's just inherent property of glass bead vesicle formation?

-> We used a peptide size similar to Huber et al. The vesicles also form without using the glass bead method and they are slightly smaller. But in this case encapsulation is not possible. Furthermore, the vesicle size also

depends a little on the EF concentration when the glass beads are used. We added data regarding this into the SI. Finally, it is an inherent property of the glass bead formation, but we also used an EF concentration and therefore a size that worked best with the TX-TL system.

The author mention that they used phosphoenolpyruvate (PEP) instead of 3-phosphoglyceric acid (3PGA) as the energy source. Was the reason for that simple the world-wide shortage of 3PGA we're currently experiencing? or was there another reason for using PEP instead of 3PGA?

It would be particularly useful to know if authors noticed any advantages of one energy system over the other in the peptide vesicles, making it differ from TxTI performance in lipid vesicles

-> Thank for this comment. We didn't know about the world-wide shortage of 3PGA, because in our lab we usually use PEP. Therefore, we didn't test for any advantages or disadvantages.

I don't like "guide to the eye" lines on figure 2. There is nothing technically wrong with it, and authors clearly stated those are not fits. Please treat this more as a comment than suggestion: in my opinion, lines on charts should only come from data and fits.

I think this is very valuable and interesting paper, and it would be of great interest to the broad readership of Nature Communications.

Reviewer #2 (Remarks to the Author):

The manuscript "Towards synthetic cells using peptide-based reaction compartments" by Vogele et al. describes the generation of peptide vesicles as potential artificial cell-like compartments. Successful transcription (RNA aptamer) and translation (fluorescent protein) inside the peptide vesicles is demonstrated. This was followed by expression of the vesicle peptide itself and evidence of incorporation of the peptide into the vesicle membrane which enabled vesicle growth from within.

This work is a major step towards the development of artificial cell compartments based on non-lipid membranes and towards the generation of protocells. The manuscript is well written but should be expanded, particularly in the introduction and discussion to give more depth to the work. Other potential applications for such peptide vesicles e.g. drug delivery should be explored. The figures are accurately presented and the claims made are novel and will be of great interest to many others in the field.

Comments:

Clearer description and more detailed graphical representation of the peptide building blocks and proposed vesicle formation is required. From Figure 1a and Figure 1b it is difficult to see how the peptide is thought to assemble. It would be useful to describe the rationale for the design of the EF peptide.

-> Thank you for this suggestion. We added details of the peptide building blocks to Figure 1a and briefly described the application of ELP and their phase transition for our design.

The authors demonstrate that the ELP vesicles are formed in vitro and provide electron microscopy images of vesicles to evidence their claim. When swelling the peptides, the vesicles reach round 180 nm in diameter. Is anything known about the size-determining factors? Is the size peptide concentration-dependent and can it be modified by changing the peptide concentration or other conditions? Have experiments carried out to test various conditions? If so, the findings should be included in the manuscript.

-> We used a peptide size similar to Huber et al. in the reference list. The vesicles also form without using the glass bead method and they are slightly smaller. But in this case encapsulation is not possible. Furthermore, the vesicle size also depends on the EF concentration when the glass beads are used. We added data regarding this into the SI. The magnitude of the size of the vesicles should also be influenced by the production methods, in this case the glass beads formation. As mentioned in the manuscript ELP exhibit a large water content in their hydrophobic state. Therefore, I doubt that a simple geometric model of the single ELP, in particular its shape, could give some insight.

ELP vesicles have been shown to assemble in vivo by Huber et. al. Can ELP vesicles self-assemble in vitro without prior adsorption onto glass beads?

-> Please see our answer to the last comment.

The authors show encapsulation of fluorescent DNA by flow cytometry as an indirect measure of encapsulation. It would be more convincing to show light microscopy images of vesicles containing the fluorescent dye. I suggest such images are added.

-> We added microscopy images of vesicles with encapsulated fluorescent DNA to the SI. Due to their small size the vesicles are only visible in the fluorescence channel and cannot colocalized in the bright field channel.

When encapsulating cell-free transcription/translation the vesicle samples contain many impurities (seen in

Figures S8-S11) and vesicles are not clearly visible for size determination. Transcription/translation outside the vesicles needed to be inhibited. Have the authors attempted to purify the “loaded” vesicles to avoid some of these issues?

-> This is a very important point. We did not purify our samples to keep the partition coefficients of involved molecules resp. the osmotic pressure constant. A change might affect the membrane stability negatively, because the equilibrium of the system would be altered.

Membrane thickness measurements are presented. If I understand correctly these measurements have been carried out on intact vesicles deposited on EM grids. However, the membrane thickness cannot be measured without sectioning through the vesicles. If the authors want to compare membrane thickness, the vesicles should be embedded in resin and sectioned through.

-> The reviewer is right. The vesicles were deposited on TEM grids and the membrane thickness was roughly determined by analyzing the TEM images. Our intention was to give only a rough estimate of the membrane thickness.

Other comments:

Introduction:

Briefly explain the TX-TL system.

-> We added a brief description of the TX-TL system and added a new reference about the used system.

“These peptides can be easily expressed in cell-free systems and thus they simplify the synthesis” Have these peptides been expressed in cell-free before?

-> ELPs indeed have been expressed in cell-free before. We added a previous publication to the references.

Results:

It is mentioned that the ELP becomes hydrophilic or hydrophobic depending on the incubation temperature. This needs to be better explained.

-> Thank you for this suggestion. We added a brief explanation to the manuscript.

The EF protein has been purified and purity confirmed by SDS gel and mass spectrometry. The mass spec data and gel should be shown in the supplementary.

-> We added the SDS gel to the SI. The mass spec data is already included.

The dBroccoli RNA aptamer needs to be better explained.

-> We added a brief description of the aptamer and a new reference to the manuscript.

Figure 3d: The reference protein (His-tagged helicase) is hardly visible on the gel.

-> Thank you for the comment. We adjusted the contrast in Fig. 3d and in the SI.

Methods:

Western blotting:

“specific anti-His antibody” Which antibody was used?

-> The primary antibody was 6x-His Epitope tag antibody (mouse, Lot:SI252938, Catalogue: MA1135). The secondary antibody was anti-mouse Alexa Fluor 680 (goat, Lot: RG233737A, Catalogue: A28183). We added this information to the main part and SI.

Supplementary:

Vesicle size distribution: S8, S9, S10, S11. In most of these images it is very difficult to see the vesicles and it is unclear what all the different images are. Are they multiple examples of the same sample?

-> We affirm the poor visibility of the vesicles in the TEM images. But we did not purify the samples from TX-TL in the surrounding solution to keep the osmotic pressure constant. As a result, we got these images with TX-TL “impurities”. Each figure shows TEM images from the same grid and therefore the same sample. We added this information to the figure captions.

S12 shows “size distribution from about 100 data points” Are you referring to the measurement of 100 vesicles?

-> Thank you for this comment. We changed the term “data points” to “vesicles”.

REVIEWERS' COMMENTS:

Reviewer #1 (Remarks to the Author):

The Authors answered all my questions and comments.

Response: We are very thankful for reviewing our manuscript.

Reviewer #2 (Remarks to the Author):

I would like to thank the authors for their responses to the comments. Looking at the point by point response letter and the revised manuscript, I think that the points raised in the previous round of review have been satisfactorily addressed.

Perhaps, some points made in the author's replies could be included in the main text of the manuscript. These are:

Where the size of the vesicles is discussed in the main text the authors could refer to their additional work now included in the supplementary subchapter 2.2 "Vesicle size depends on EF concentration". Also, the peptide concentration used for vesicle formation is mentioned in the supplementary but seems to be missing in the main text.

Response: Thank you for this suggestion. We included the EF concentrations and the corresponding vesicle sizes in the main manuscript.

The rationale for the chosen peptide concentration could be better explained.

Supplementary, subchapter 2.2 "For further experiments we used a concentration of 180 pM since it worked well with TX-TL"

I am not clear what the authors mean here. Where the TX-TL components most efficiently encapsulated at 180 pM EF? What were the issues with vesicles formed from the other tested EF concentrations?

Response: Our apologies for this unclear statement. 180 pM EF showed the lowest relative dispersion (see revised manuscript). When 180 pM EF were used the expression levels were slightly higher compared to the other concentrations, but not significantly.

The authors added a microscopy bright field and a fluorescence microscopy image of the vesicles to subchapter 2.2 in the supplementary. This could be referred to in the main text.

Response: We included the reference.

It would be useful to explain in the text why the vesicles were not attempted to be purified. It has been explained in the response letter but the text was not included in the manuscript.

Response: I am very thankful for this suggestion. We included this point at the beginning of the discussion so that readers do not miss this important point.

Finally, in my opinion, this work is exciting and novel and I would very much like to see it being published.

We are very thankful for your review.